# DeViSE: A Deep Visual-Semantic Embedding Model

**Andrea Frome*, Greg S. Corrado*, Jonathon Shlens*, Samy Bengio**
**Jeffrey Dean, Marc'Aurelio Ranzato, Tomas Mikolov**

{afrome, gcorrado, shlens, bengio, jeff, ranzato[†], tmikolov}@google.com
Google, Inc.
Mountain View, CA, USA

## Abstract

Modern visual recognition systems are often limited in their ability to scale to large numbers of object categories. This limitation is in part due to the increasing difficulty of acquiring sufficient training data in the form of labeled images as the number of object categories grows. One remedy is to leverage data from other sources – such as text data – both to train visual models and to constrain their predictions. In this paper we present a new *deep visual-semantic embedding model* trained to identify visual objects using both labeled image data as well as semantic information gleaned from unannotated text. We demonstrate that this model matches state-of-the-art performance on the 1000-class ImageNet object recognition challenge while making more semantically reasonable errors, and also show that the semantic information can be exploited to make predictions about tens of thousands of image labels not observed during training. Semantic knowledge improves such *zero-shot* predictions achieving hit rates of up to $18\%$ across thousands of novel labels never seen by the visual model.

## 1 Introduction

The visual world is populated with a vast number of objects, the most appropriate labeling of which is often ambiguous, task specific, or admits multiple equally correct answers. Yet state-of-the-art vision systems attempt to solve recognition tasks by artificially assigning images to a small number of rigidly defined classes. This has led to building labeled image data sets according to these artificial categories and in turn to building visual recognition systems based on N-way discrete classifiers. While growing the number of labels and labeled images has improved the utility of visual recognition systems [7], scaling such systems beyond a limited number of discrete categories remains an unsolved problem. This problem is exacerbated by the fact that N-way discrete classifiers treat all labels as disconnected and unrelated, resulting in visual recognition systems that cannot transfer semantic information about learned labels to unseen words or phrases. One way of dealing with this issue is to respect the natural continuity of visual space instead of artificially partitioning it into disjoint categories [20].

We propose an approach that addresses these shortcomings by training a visual recognition model with both labeled images and a comparatively large and independent dataset – semantic information from unannotated text data. This *deep visual-semantic embedding* model (DeViSE) leverages textual data to learn semantic relationships between labels, and explicitly maps images into a rich semantic embedding space. We show that this model performs comparably to state-of-the-art visual object classifiers when trained and evaluated on flat 1-of-N metrics, while simultaneously making fewer semantically unreasonable mistakes along the way. Furthermore, we show that the model leverages

---
[†]Current affiliation: Facebook, Inc.

visual and semantic similarity to correctly predict object category labels for unseen categories, i.e. "zero-shot" classification, even when the number of unseen visual categories is 20,000 for a model trained on just 1,000 categories.

## 2    Previous Work

The current state-of-the-art approach to image classification is a deep convolutional neural network trained with a softmax output layer (i.e. multinomial logistic regression) that has as many units as the number of classes (see, for instance [11]). However, as the number of classes grows, the distinction between classes blurs, and it becomes increasingly difficult to obtain sufficient numbers of training images for rare concepts.

One solution to this problem, termed WSABIE [20], is to train a joint embedding model of both images and labels, by employing an online learning-to-rank algorithm. The proposed model contained two sets of parameters: (1) a linear mapping from image features to the joint embedding space, and (2) an embedding vector for each possible label. Compared to the proposed approach, WSABIE only explored linear mappings from image features to the embedding space, and the available labels were only those provided in the image training set. It could thus not generalize to new classes.

More recently, Socher et al [18] presented a model for *zero-shot learning* where a deep neural network was first trained in an unsupervised manner from many images in order to obtain a rich image representation [3]; in parallel, a neural network language model [2] was trained in order to obtain embedding representations for thousands of common terms. The authors trained a linear mapping between the image representations and the word embeddings representing 8 classes for which they had labeled images, thus linking the image representation space to the embedding space. This last step was performed using a mean-squared error criterion. They also trained a simple model to determine if a given image was from any of the 8 original classes or not (i.e., an outlier detector). When the model determined an image to be in the set of 8 classes, a separately trained softmax model was used to perform the 8-way classification; otherwise the model predicted the nearest class in the embedding space (in their setting, only 2 outlier classes were considered). Their model differs from our proposed approach in several ways: first and foremost, the scale, as our model considers 1,000 known classes for the image model and up to 20,000 unknown classes, instead of respectively 8 and 2; second, in [18] there is an inherent trade-off between the quality of predictions for trained and outlier classes; third, by using a different visual model, different language model, and different training objective, we were able to train a single unified model that uses only embeddings.

There has been other recent work showing impressive zero-shot performance on visual recognition tasks [12, 17, 16], however all of these rely on a curated source of semantic information for the labels: the WordNet hierarchy is used in [12] and [17], and [16] uses a knowledge base containing descriptive properties for each class. By contrast, our approach learns its semantic representation directly from unannotated data.

## 3    Proposed Approach

Our objective is to leverage semantic knowledge learned in the text domain, and transfer it to a model trained for visual object recognition. We begin by pre-training a simple neural language model well-suited for learning semantically-meaningful, dense vector representations of words [13]. In parallel, we pre-train a state-of-the-art deep neural network for visual object recognition [11], complete with a traditional softmax output layer. We then construct a deep visual-semantic model by taking the lower layers of the pre-trained visual object recognition network and re-training them to predict the vector representation of the image label text as learned by the language model. These three training phases are detailed below.

### 3.1    Language Model Pre-training

The skip-gram text modeling architecture introduced by Mikolov et al [13, 14] has been shown to efficiently learn semantically-meaningful floating point representations of terms from unannotated text. The model learns to represent each term as a fixed length embedding vector by predicting adjacent terms in the document (Figure 1a, right). We call these vector representations *embedding*

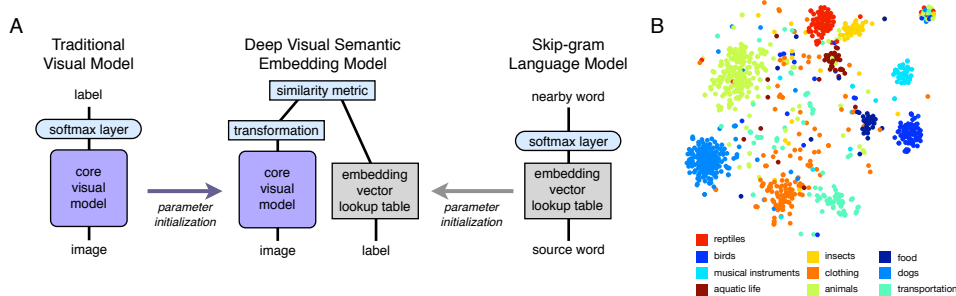

Figure 1: (a) Left: a visual object categorization network with a softmax output layer; Right: a skip-gram language model; Center: our joint model, which is initialized with parameters pre-trained at the lower layers of the other two models. (b) t-SNE visualization [19] of a subset of the ILSVRC 2012 1K label embeddings learned using skip-gram.

*vectors*. Because synonyms tend to appear in similar contexts, this simple objective function drives the model to learn similar embedding vectors for semantically related words.

We trained a skip-gram text model on a corpus of 5.7 million documents (5.4 billion words) extracted from `wikipedia.org`. The text of the web pages was tokenized into a lexicon of roughly 155,000 single- and multi-word terms consisting of common English words and phrases as well as terms from commonly used visual object recognition datasets [7]. Our skip-gram model used a hierarchical softmax layer for predicting adjacent terms and was trained using a 20-word window with a single pass through the corpus. For more details and a pointer to open-source code, see [13].

We trained skip-gram models of varying hidden dimensions, ranging from 100-D to 2,000-D, and found 500- and 1,000-D embeddings to be a good compromise between training speed, semantic quality, and the ultimate performance of the DeViSE model described below. The semantic quality of the embedding representations learned by these models is impressive.[1] A visualization of the language embedding space over a subset of ImageNet labels indicates that the language model learned a rich semantic structure that could be exploited in vision tasks (Figure 1b).

## 3.2  Visual Model Pre-training

The visual model architecture we employ is based on the winning model for the 1,000-class ImageNet Large Scale Visual Recognition Challenge (ILSVRC) 2012 [11, 6]. The deep neural network model consists of several convolutional filtering, local contrast normalization, and max-pooling layers, followed by several fully connected neural network layers trained using the dropout regularization technique [10]. We trained this model with a softmax output layer, as described in [11], to predict one of 1,000 object categories from the ILSVRC 2012 1K dataset [7], and were able to reproduce their results. This trained model serves both as our benchmark for performance comparisons, as well as the initialization for our joint model.

## 3.3  Deep Visual-Semantic Embedding Model

Our deep visual-semantic embedding model (DeViSE) is initialized from these two pre-trained neural network models (Figure 1a). The embedding vectors learned by the language model are unit normed and used to map label terms into target vector representations[2].

The core visual model, with its softmax prediction layer now removed, is trained to predict these vectors for each image, by means of a projection layer and a similarity metric. The projection layer is a linear transformation that maps the 4,096-D representation at the top of our core visual model into the 500- or 1,000-D representation native to our language model.

The choice of loss function proved to be important. We used a combination of dot-product similarity and hinge rank loss (similar to [20]) such that the model was trained to produce a higher dot-product similarity between the visual model output and the vector representation of the correct label than between the visual output and other randomly chosen text terms. We defined the per training example hinge rank loss:

$$loss(image, label) = \sum_{j \neq label} \max[0, margin - \vec{t}_{label} M \vec{v}(image) + \vec{t}_j M \vec{v}(image)] \qquad (1)$$

where $\vec{v}(image)$ is a column vector denoting the output of the top layer of our core visual network for the given image, $M$ is the matrix of trainable parameters in the linear transformation layer, $\vec{t}_{label}$ is a row vector denoting learned embedding vector for the provided text label, and $\vec{t}_j$ are the embeddings of other text terms. In practice, we found that it was expedient to randomize the algorithm both by (1) restricting the set of false text terms to possible image labels, and (2) truncating the sum after the first margin-violating false term was encountered. The $\vec{t}$ vectors were constrained to be unit norm, and a fixed $margin$ of 0.1 was used in all experiments[3]. We also experimented with an $L_2$ loss between visual and label embeddings, as suggested by Socher et al. [18], but that consistently yielded about half the accuracy of the rank loss model. We believe this is because the nearest neighbor evaluation is fundamentally a ranking problem and is best solved with a ranking loss, whereas the $L_2$ loss only aims to make the vectors close to one another but remains agnostic to incorrect labels that are closer to the target image.

The DeViSE model was trained by asynchronous stochastic gradient descent on a distributed computing platform described in [4]. As above, the model was presented only with images drawn from the ILSVRC 2012 1K training set, but now trained to predict the term strings as text[4]. The parameters of the projection layer $M$ were first trained while holding both the core visual model and the text representation fixed. In the later stages of training the derivative of the loss function was back-propagated into the core visual model to fine-tune its output[5], which typically improved accuracy by 1-3% (absolute). Adagrad per-parameter dynamic learning rates were utilized to keep gradients well scaled at the different layers of the network [9].

At test time, when a new image arrives, one first computes its vector representation using the visual model and the transformation layer; then one needs to look for the nearest labels in the embedding space. This last step can be done efficiently using either a tree or a hashing technique, in order to be faster than the naive linear search approach (see for instance [1]). The nearest labels are then mapped back to ImageNet synsets for scoring (see Supplementary Materials for details).

## 4  Results

The goals of this work are to develop a vision model that makes semantically relevant predictions even when it makes errors and that generalizes to classes outside of its labeled training set, i.e. *zero-shot learning*. We compare DeViSE to two models that employ the same high-quality core vision model, but lack the semantic structure imparted by our language model: (1) a *softmax baseline* model – a state-of-the-art vision model [11] which employs a 1000-way softmax classifier; (2) a *random embedding* model – a version of our model that uses random unit-norm embedding vectors in place of those learned by the language model. Both use the trained visual model described in Section 3.2.

In order to demonstrate parity with the softmax baseline on the most commonly-reported metric, we compute "flat" hit@$k$ metrics – the percentage of test images for which the model returns the one true label in its top $k$ predictions. To measure the semantic quality of predictions beyond the true label, we employ a hierarchical precision@$k$ metric based on the label hierarchy provided with the

| Model type | dim | Flat hit@$k$ (%) | | | | Hierarchical precision@$k$ | | | |
|---|---|---|---|---|---|---|---|---|---|
| | | 1 | 2 | 5 | 10 | 2 | 5 | 10 | 20 |
| Softmax baseline | N/A | **55.6** | **67.4** | **78.5** | **85.0** | 0.452 | 0.342 | 0.313 | 0.319 |
| DeViSE | 500 | 53.2 | 65.2 | 76.7 | 83.3 | 0.447 | **0.352** | **0.331** | **0.341** |
| | 1000 | 54.9 | 66.9 | 78.4 | **85.0** | **0.454** | 0.351 | 0.325 | 0.331 |
| Random embeddings | 500 | 52.4 | 63.9 | 74.8 | 80.6 | 0.428 | 0.315 | 0.271 | 0.248 |
| | 1000 | 50.5 | 62.2 | 74.2 | 81.5 | 0.418 | 0.318 | 0.290 | 0.292 |
| Chance | N/A | 0.1 | 0.2 | 0.5 | 1.0 | 0.007 | 0.013 | 0.022 | 0.042 |

Table 1: Comparison of model performance on our test set, taken from the ImageNet ILSVRC 2012 1K validation set. Note that hierarchical precision@1 is equivalent to flat hit@1. See text for details.

ImageNet image repository [7]. In particular, for each true label and value of $k$, we generate a ground truth list from the semantic hierarchy, and compute a per-example precision equal to the fraction of the model's $k$ predictions that overlap with the ground truth list. We report mean precision across the test set. Detailed descriptions of the generation of the ground truth lists, the hierarchical scoring metric, and train/validation/test dataset splits are provided in the Supplementary Materials.

## 4.1 ImageNet (ILSVRC) 2012 1K Results

This section presents flat and hierarchical results on the ILSVRC 2012 1K dataset, where the classes of the examples presented at test time are the same as those used for training. Table 1 shows results for the DeViSE model for 500- and 1000-dimensional skip-gram models compared to the random embedding and softmax baseline models, on both the flat and hierarchical metrics.[6]

On the flat metric, the softmax baseline shows higher accuracy for $k = 1, 2$. At $k = 5, 10$, the 1000-D DeViSE model has reached parity, and at $k = 20$ (not shown) it performs slightly better. We expected the softmax model to be the best performing model on the flat metric, given that its cross-entropy training objective is most well matched to the evaluation metric, and are surprised that the performance of DeViSE is so close to softmax performance.

On the hierarchical metric, the DeViSE models show better semantic generalization than the softmax baseline, especially for larger $k$. At $k = 5$, the 500-D DeViSE model shows a 3% relative improvement over the softmax baseline, and at $k = 20$ almost a 7% relative improvement. This is a surprisingly large gain, considering that the softmax baseline is a reproduction of the best published model on these data. The gap that exists between the DeViSE model and softmax baseline on the hierarchical metric reflects the benefit of semantic information above and beyond visual similarity [8]. The gap between the DeViSE model and the random embeddings model establishes that the source of the gain is the well-structured embeddings learned by the language model not some other property of our architecture.

## 4.2 Generalization and Zero-Shot Learning

A distinct advantage of our model is its ability to make reasonable inferences about candidate labels it has never visually observed. For example, a DeViSE model trained on images labeled *tiger shark*, *bull shark*, and *blue shark*, but never with images labeled *shark*, would likely have the ability to generalize to this more coarse-grained descriptor because the language model has learned a representation of the general concept of *shark* which is similar to all of the specific sharks. Similarly, if tested on images of highly specific classes which the model has never seen before, for example a photo of an oceanic whitecap shark, and asked whether the correct label is more likely *oceanic whitecap shark* or some other unfamiliar label (say, *nuclear submarine*), our model stands a fighting chance of guessing correctly because the language model ensures that representation of *oceanic whitecap shark* is closer to the representation of sharks the model *has* seen, while the representation of *nuclear submarine* is closer to those of other sea vessels.

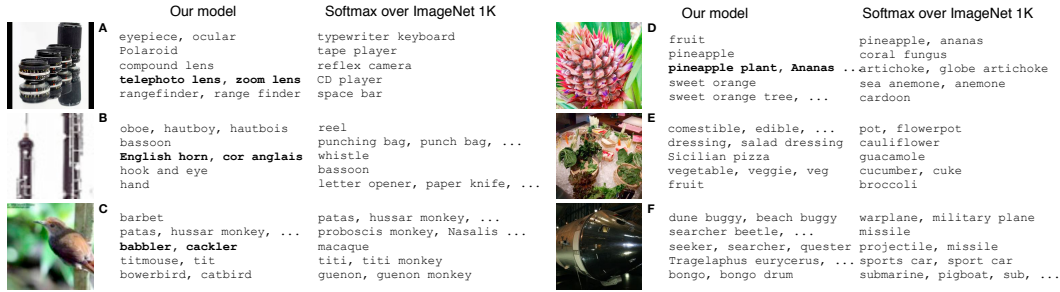

| | Our model | Softmax over ImageNet 1K | | Our model | Softmax over ImageNet 1K |
|---|---|---|---|---|---|
| **A** | eyepiece, ocular<br>Polaroid<br>compound lens<br>**telephoto lens, zoom lens**<br>rangefinder, range finder | typewriter keyboard<br>tape player<br>reflex camera<br>CD player<br>space bar | **D** | fruit<br>pineapple<br>**pineapple plant, Ananas** ..<br>sweet orange<br>sweet orange tree, ... | pineapple, ananas<br>coral fungus<br>artichoke, globe artichoke<br>sea anemone, anemone<br>cardoon |
| **B** | oboe, hautboy, hautbois<br>bassoon<br>**English horn, cor anglais**<br>hook and eye<br>hand | reel<br>punching bag, punch bag, ...<br>whistle<br>bassoon<br>letter opener, paper knife, ... | **E** | comestible, edible, ...<br>dressing, salad dressing<br>Sicilian pizza<br>vegetable, veggie, veg<br>fruit | pot, flowerpot<br>cauliflower<br>guacamole<br>cucumber, cuke<br>broccoli |
| **C** | barbet<br>patas, hussar monkey, ...<br>**babbler, cackler**<br>titmouse, tit<br>bowerbird, catbird | patas, hussar monkey, ...<br>proboscis monkey, Nasalis ...<br>macaque<br>titi, titi monkey<br>guenon, guenon monkey | **F** | dune buggy, beach buggy<br>searcher beetle, ...<br>seeker, searcher, quester<br>Tragelaphus eurycerus, ...<br>bongo, bongo drum | warplane, military plane<br>missile<br>projectile, missile<br>sports car, sport car<br>submarine, pigboat, sub, ... |

Figure 2: For each image, the top 5 zero-shot predictions of DeViSE+1K from the 2011 21K label set and the softmax baseline model, both trained on ILSVRC 2012 1K. Predictions ordered by decreasing score, with correct predictions in bold. Ground truth: (a) *telephoto lens, zoom lens*; (b) *English horn, cor anglais*; (c) *babbler, cackler*; (d) *pineapple, pineapple plant, Ananas comosus*; (e) *salad bar*; (f) *spacecraft, ballistic capsule, space vehicle*.

| | | | Flat hit@$k$ (%) | | | | |
|---|---|---|---|---|---|---|---|
| Data Set | Model | # Candidate Labels | 1 | 2 | 5 | 10 | 20 |
| 2-hop | DeViSE-0 | 1,589 | 6.0 | 10.0 | 18.1 | 26.4 | 36.4 |
| | DeViSE+1K | 2,589 | 0.8 | 2.7 | 7.9 | 14.2 | 22.7 |
| 3-hop | DeViSE-0 | 7,860 | 1.7 | 2.9 | 5.3 | 8.2 | 12.5 |
| | DeViSE+1K | 8,860 | 0.5 | 1.4 | 3.4 | 5.9 | 9.7 |
| ImageNet 2011 21K | DeViSE-0 | 20,841 | 0.8 | 1.4 | 2.5 | 3.9 | 6.0 |
| | DeViSE+1K | 21,841 | 0.3 | 0.8 | 1.9 | 3.2 | 5.3 |

Table 2: Flat hit@$k$ performance of DeViSE on ImageNet-based zero-shot datasets of increasing difficulty from top to bottom. DeViSE-0 and DeViSE+1K are the same trained model, but DeViSE-0 is restricted to only predict zero-shot classes, whereas DeViSE+1K predicts both the zero-shot and the 1K training labels. For all, zero-shot classes did not occur in the image training set.

To test this hypothesis, we extracted images from the ImageNet 2011 21K dataset with labels that were not included in the ILSVRC 2012 1K dataset on which DeViSE was trained. These are "zero-shot" data sets in the sense that our model has no visual knowledge of these labels, though embeddings for the labels were learned by the language model. The softmax baseline is only able to predict labels from ILSVRC 2012 1K. The zero-shot experiments were performed with the same trained 500-D DeViSE model used for results in Section 4.1, but it is evaluated in two ways: DeViSE-0 only predicts the zero-shot labels, and DeViSE+1K predicts zero-shot labels and the ILSVRC 2012 1K training labels.

Figure 2 shows label predictions for a handful of selected examples from this dataset to qualitatively illustrate model behavior. Note that DeViSE successfully predicts a wide range of labels outside its training set, and furthermore, the incorrect predictions are generally semantically "close" to the desired label. Figure 2 (a), (b), (c), and (d) show cases where our model makes significantly better top-5 predictions than the softmax-based model. For example, in Figure 2 (a), the DeViSE model is able to predict a number of lens-related labels even though it was not trained on images in any of the predicted categories. Figure 2 (d) illustrates a case where the top softmax prediction is quite good, but where it is unable to generalize to new labels and its remaining predictions are off the mark, while our model's predictions are more plausible. Figure 2 (e) highlights a case where neither model gets the exact true label, but both models are giving plausible labels. Figure 2 (f) shows a case where the softmax model emits more nearly correct labels than the DeViSE model.

To quantify the performance of the model on zero-shot data, we constructed from our ImageNet 2011 21K zero-shot data three test data sets of increasing difficulty based on the image labels' tree distance from the training ILSVRC 2012 1K labels in the ImageNet label hierarchy [7]. The easiest dataset, "2-hop", is comprised of the 1,589 labels that are within two tree hops of the training labels, making them visually and semantically similar to the training set. A more difficult "3-hop" dataset was constructed in the same manner. Finally, we built a third, particularly challenging dataset consisting of all the labels in ImageNet 2011 21K that are not in ILSVRC 2012 1K.

| Data Set | Model | Hierarchical precision@$k$ | | | | |
|---|---|---|---|---|---|---|
| | | 1 | 2 | 5 | 10 | 20 |
| 2-hop | DeViSE-0 | **0.06** | 0.152 | 0.192 | **0.217** | **0.233** |
| | DeViSE+1K | 0.008 | 0.204 | **0.196** | 0.201 | 0.214 |
| | Softmax baseline | 0 | **0.236** | 0.181 | 0.174 | 0.179 |
| 3-hop | DeViSE-0 | **0.017** | 0.037 | 0.191 | **0.214** | **0.236** |
| | DeViSE+1K | 0.005 | **0.053** | **0.192** | 0.201 | 0.214 |
| | Softmax baseline | 0 | 0.053 | 0.157 | 0.143 | 0.130 |
| ImageNet 2011 21K | DeViSE-0 | **0.008** | 0.017 | 0.072 | 0.085 | 0.096 |
| | DeViSE+1K | 0.003 | **0.025** | **0.083** | **0.092** | **0.101** |
| | Softmax baseline | 0 | 0.023 | 0.071 | 0.069 | 0.065 |

Table 3: Hierarchical precision@$k$ results on zero-shot classification. Performance of DeViSE compared to the softmax baseline model across the same datasets as in Table 2. Note that the softmax model can never directly predict the correct label so its precision@1 is 0.

| Model | 200 labels | 1000 labels |
|---|---|---|
| DeViSE | 31.8% | 9.0% |
| Mensink et al. 2012 [12] | 35.7% | 1.9% |
| Rohrbach et al. 2011 [17] | 34.8% | - |

Table 4: Flat hit@5 accuracy on the zero-shot task from [12]. DeViSE experiments were performed with a 500-D model. The [12] model uses a curated hierarchy over labels for zero-shot classification, but without using this information, our model is close in performance on the 200 zero-shot class label task. When the models can predict any of the 1000 labels, we achieve better accuracy, indicating DeViSE has less of a bias toward training classes than [12]. As in [12], we include a result on a similar task from [17], though their work used a different set of 200 zero-shot classes.

We again calculated the flat hit@$k$ measure to determine how frequently DeViSE-0 and DeViSE+1K predicted the correct label for each of these data sets (Table 2). DeViSE-0's top prediction was the correct label 6.0% of the time across 1,589 novel labels, and the rate increases with $k$ to 36.4% within the top 20 predictions. As the zero-shot data sets become more difficult, the accuracy decreases in absolute terms, though it is better relative to chance (not shown). Since a traditional softmax visual model can *never* produce the correct label on zero-shot data, its performance would be 0% for all $k$. The DeViSE+1K model performed uniformly worse than the plain DeViSE-0 model by a margin that indicates it has a bias toward training classes.

To provide a stronger baseline for comparison, we compared the performance of our model and the softmax model on the hierarchical metric we employed above. Although the softmax baseline model can never predict exactly the correct label, the hierarchical metric will give the model credit for predicting labels that are in the neighborhood of the correct label in the ImageNet hierarchy (for $k > 1$). Visual similarity is strongly correlated with semantic similarity for nearby object categories [8], and the softmax model does leverage visual similarity between zero-shot and training images to make predictions that will be scored favorably (e.g. Figure 2d).

The easiest dataset, "2-hop", contains object categories that are as visually and semantically similar to the training set as possible. For this dataset the softmax model outperforms the DeViSE model for hierarchical precision@2, demonstrating just how large a role visual similarity plays in predicting semantically "nearby" labels (Table 3). However, for $k = 5, 10, 20$, our model produces superior predictions relative to the ImageNet hierarchy, even on this easiest dataset. For the two more difficult datasets, where there are more novel categories and the novel categories are less closely related to those in the training data set, DeViSE outperforms the softmax model at all measured hierarchical precisions. The quantitative gains can be quite large, as much as 82% relative improvement over softmax performance, and qualitatively, the softmax model's predictions can be surprisingly unreasonable in some cases (e.g. Figure 2c). The random embeddings model we described above performed substantially worse than either of the real models. These results indicate that our architecture succeeds in leveraging the semantic knowledge captured by the language model to make reasonable predictions, even as test images become increasingly dissimilar from those used in the training set.

To provide a comparison with other work in zero-shot learning, we also directly compare to the zero-shot results from [12]. These were performed on a particular 800/200 split of the 1000 classes

from ImageNet 2010: training and model tuning is performed using the 800 classes, and test images are drawn from the remaining 200 classes. Results are shown in Table 4.

Taken together, these zero-shot experiments indicate that the DeViSE model can exploit both visual and semantic information to predict novel classes never before observed. Furthermore, the presence of semantic information in the model substantially improves the quality of its predictions.

## 5    Conclusion

In contrast to previous attempts in this area [18], we have shown that our joint visual-semantic embedding model can be trained to give performance comparable to a state-of-the-art softmax based model on a flat object classification metric, while simultaneously making more semantically reasonable errors, as indicated by its improved performance on a hierarchical label metric. We have also shown that this model is able to make correct predictions across thousands of previously unseen classes by leveraging semantic knowledge elicited only from unannotated text.

The advantages of this architecture, however, extend beyond the experiments presented here.

First, we believe that our model's unusual compatibility with larger, less manicured data sets will prove to be a major strength moving forward. In particular, the skip-gram language model we constructed included only a modestly sized vocabulary, and was exposed only to the text of a single online encyclopedia; we believe that the gains available to models with larger vocabularies and trained on vastly larger text corpora will be significant, and easily outstrip methods which rely on manually constructed semantic hierarchies (e.g. [17]). Perhaps more importantly, though here we trained on a curated academic image dataset, our model's architecture naturally lends itself to being trained on all available images that can be annotated with any text term contained in the (larger) vocabulary. We believe that training massive "open" image datasets of this form will dramatically improve the quality of visual object categorization systems.

Second, we believe that the 1-of-N (and nearly balanced) visual object classification problem is soon to be outmoded by practical visual object categorization systems that can handle very large numbers of labels [5] and the re-definition of valid label sets at test time. For example, our model can be trained once on all available data, and simultaneously used in one application requiring only coarse object categorization (e.g. house, car, pedestrian) and another application requiring fine categorization in a very specialized subset (e.g. Honda Civic, Ferrari F355, Tesla Model-S). Moreover, because test time computation can be sub-linear in the number of labels contained in the training set, our model can be used in exactly such systems with much larger numbers of labels, including overlapping or never-observed categories.

Moving forward, we are experimenting with techniques which more directly leverage the structure inherent in the learned language embedding, greatly reducing training costs of the joint model and allowing even greater scaling [15].

**Acknowledgments**

Special thanks to those who lent their insight and technical support for this work, including Matthieu Devin, Alex Krizhevsky, Quoc Le, Rajat Monga, Ilya Sutskever, and Wojciech Zaremba.

## Footnotes

[1]For example, the 9 nearest terms to *tiger shark* using cosine distance are *bull shark*, *blacktip shark*, *shark*, *oceanic whitetip shark*, *sandbar shark*, *dusky shark*, *blue shark*, *requiem shark*, and *great white shark*. The 9 nearest terms to *car* are *cars*, *muscle car*, *sports car*, *compact car*, *automobile*, *racing car*, *pickup truck*, *dealership*, and *sedans*.

[2]In [13], which introduced the skip-gram model for text, cosine similarity between vectors is used for measuring semantic similarity. Unit-norming the vectors and using dot product similarity is an equivalent similarity measurement.

[3]The margin was chosen to be a fraction of the norm of the vectors, which is 1.0. A wide range of values would likely work well.

[4]ImageNet image labels are *synsets*, a set of synonymous terms, where each term is a word or phrase. We found training the model to predict the first term in each synset to be sufficient, but sampling from the synset terms might work equally well.

[5]In principle the gradients can also be back-propagated into the vector representations of the text labels. In this case, the language model should continue to train simultaneously in order to maintain the global semantic structure over all terms in the vocabulary.

[6]Note that our softmax baseline results differ from the results in [11] due to a simplification in the evaluation procedure: [11] creates several distorted versions of each test image and aggregates the results for a final label, whereas in our experiments, we evaluate using only the original test image. Our softmax baseline is able to reproduce the performance of the model in [11] when evaluated with the same procedure.

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
