[Supplementary Material · appendix_main.pdf]

# DeViSE: A Deep Visual-Semantic Embedding Model: Appendix

**Andrea Frome\*, Greg S. Corrado\*, Jonathon Shlens\*, Samy Bengio**
**Jeffrey Dean, Marc'Aurelio Ranzato, Tomas Mikolov**

{afrome, gcorrado, shlens, bengio, jeff, ranzato†, tmikolov}@google.com
Google, Inc.
Mountain View, CA, USA

## A    Appendix

### A.1    Validation/test Data Methodology

For all experiments except the comparison to Mensink et al. [12] we train our visual model and DeViSE on the ILSVRC 2012 1K data set. Experiments in Section 4.1 use images and labels only from the 2012 1K set for testing, and the zero-shot experiments in Section 4.2 use image and labels from the ImageNet 2011 21K set and, for DeViSE+1K, also labels from ILSVRC 2012 1K. The same subset of the ILSVRC 2012 1K data used to train the visual model (Section 3.2) is also used to train DeViSE and the random embedding-based models, and when training all models, we randomly flip, crop and translate images to artificially expand the training set several-fold. The 50K images from the ILSVRC 2012 1K validation set are split randomly 10/90 into our experimental validation and held-out test sets of 5K and 45K images, respectively. Our validation set is used to choose hyperparameters, and results are reported on the held-out set. The 1,000 classes are roughly balanced in the validation and held-out sets. The 500-D DeViSE model trained for the experiments in Section 4.1 is also used for the corresponding zero-shot experiments in Section 4.2 with no additional tuning.

The zero-shot experiments performed to compare to Mensink et al. [12] are trained with images and labels from the ILSVRC 2010 1K data set, using the same 800/200 training/test class split used in [12]. We use the ILSVRC 2010 training/validation/test data split; training and validation images from the 800 classes are used to train and tune our visual model and DeViSE, and test images from the 200 zero-shot classes are used to generate our experimental results.

At test time, images are center-cropped to $225 \times 225$ for input to the visual model, and no other distortions are applied.

### A.2    Mapping Text Terms to ImageNet Synset

The language model represents terms and phrases gathered from unannotated text as embedding vectors, whereas the ImageNet data set represents each class as a *synset*, a set of synonymous terms, where each term is a word or phrase. When training DeViSE, a method is needed for mapping from an ImageNet synset to the target embedding vector, and at prediction time, label predictions from the embedding space must be translated back into ImageNet synsets from the test set for scoring. There are two complications: (1) the same term can occur in multiple ImageNet synsets, often representing different concepts, for example the two synsets consisting only of "crane" in ILSVRC 2012 1K; (2) the language model as we have trained it only has one embedding for each word or phrase, so there is only one embedding vector representing both senses of "crane".

When training, we choose the target embedding vector by mapping the first term of the synset to its embedding vector through the text of the synset term. We found this to work well in practice; other possible approaches are to choose randomly from among the synset terms or take an average of the embedding vectors for the synset terms.

When making a prediction, the mapping from embedded text vectors back to ImageNet synsets is more difficult: each embedding vector can correspond to several ImageNet synsets due to the repetition of terms between synsets, up to 9 different synsets in the case of "head". Additionally, multiple of our predicted embedding vectors can correspond to terms from the same synset, e.g. "ballroom", "dance hall", "dance palace". In practice, this happens frequently since synonymous terms are embedded close to one another by the skip-gram language model.

For a given visual output vector, our model first finds the $N$ nearest embedding label vectors using cosine similarity, sorted by their similarity, where $N > k$. In these experiments, we chose $N = 4 * k$ as this is close to the average number of text labels per synset in the ILSVRC 2012 data set. The first step in converting these to ImageNet synsets is to expand every embedded term to all of its corresponding ImageNet synsets. For example, "crane" would be expanded to the two synsets which contain the term "crane" (the order of the two "crane" synsets in the final list is arbitrary). After expansion, if there are duplicate entries for a given synset, then all but the first are removed from their places in the list, leaving a list where each synset occurs at most once in the prediction list. Finally, the list is truncated to the top $k$ predictions. We experimented with choosing randomly from among all the possible synsets instead of expanding to all of them and found this to slightly reduce performance in the ILSVRC 2012 1K experiments.

### A.3  Hierarchical Precision-at-k Metric

We defined the following hierarchical precision-at-$k$ metric, $hp@k$, to assess the accuracy of model predictions with respect to the ImageNet object category hierarchy. For each image in the test set, the model in question emits its top $k$ predicted ImageNet object category labels (synsets). We calculate $hp@k$ as the fraction of these $k$ predictions which are in $hCorrectSet$, averaged across the test examples:

$$hp@k = \frac{1}{N} \sum_{i=1}^{N} \frac{\text{number of model's top } k \text{ predictions in } hCorrectSet \text{ for image } i}{k} \qquad (1)$$

The $hCorrectSet$ for a true label is constructed by iteratively adding nodes from the ImageNet hierarchy in a widening radius around the true label until $hCorrectSet$ has a size $\geq k$:

```
hCorrectSet = {}
R = 0
while NumberElements(hCorrectSet < k):
    radiusSet = all nodes in the ImageNet hierarchy which are
                R hops from the true label
    validRadiusSet = ValidLabelNodes(radiusSet)
    hCorrectSet = Union(hCorrectSet, validRadiusSet)
    R = R + 1
return hCorrectSet
```

The size of $hCorrectSet$ for a given test example depends on the combination of the structure of the hierarchy around a given label and which classes are included in the test set. It is exactly 1 when $k = 1$ and is rarely equal to $k$ when $k > 1$. The `ValidLabelNodes()` function allows us to restrict $hCorrectSet$ to any subset of labels in the ImageNet (or larger WordNet) hierarchy. For example, in generating the results in Table 2 we restrict the nodes in the $hCorrectSet$ to be drawn from only those nodes which are both members the ImageNet 2011 21K label set *and* are three-hops or less from at least one of the ImageNet 2012 1K labels.

Note that this hierarchical metric differs from the hierarchical metric used in some of the earlier ImageNet Challenge competitions. That metric was generally considered to be rather insensitive, and was withdrawn from more recent years of the competition. Our DeViSE model does perform better than the baseline softmax model on that metric as well, but effect sizes are generally much smaller.

|  |  | $k$ | | | |
| Label set | # $kCorrectSet$ Labels | 2 | 5 | 10 | 20 |
|---|---|---|---|---|---|
| ImageNet 2012 1K | 1000 | 6.5 | 12.5 | 22.5 | 41.7 |
| Zero-shot 2-hop | 2589 | 3.2 | 16.8 | 25.5 | 45.2 |
| Zero-shot 3-hop | 8860 | 4.4 | 577.4 | 635.4 | 668.0 |
| Zero-shot ImageNet 2011 21K | 21900 | 5.4 | 273.3 | 317.4 | 350.6 |

Table 1: Mean sizes of $hCorrectSet$ lists used for hierarchical evaluation, averaged across the test examples, shown for various label sets and values of $k$. At $k = 1$, $hCorrectSet$ always contains only the true label. Note that for the zero-shot data sets, $kCorrectSet$ includes the test set labels as well as the ImageNet 2012 1K labels. List sizes vary among test examples depending upon the local topology of the graph around the true label as well as how many labels from the graph are in the ground truth set.

## Footnotes

†Current affiliation: Facebook, Inc.