[Reviews · NeurIPS 2013]

Submitted by Assigned_Reviewer_7

Paper 1048 proposes a system for large-scale zero-shot visual recognition.
It consists of the following steps:
(1) Learn an embedding of a large number of words in a Euclidean space.
(2) Learn a deep architecture which takes images as input and predicts one of 1,000 object categories.
The 1,000 categories are a subset of the 'large number of words' of step (1).
(3) Remove the last layer of the visual model -- leaving what is referred to as the 'core' visual model.
Replace it by the word embeddings and add a layer to map the core visual model output to the word embeddings.

On the positive side:
+ The paper is well written and reads easily.
+ The problem of large-scale zero shot recognition is one of high practical and scientific value.
+ The experiments are comprehensive and well-designed.
+ The paper reports state-of-the-art results on a very large scale.

On the negative side:

- The technical contribution feels somewhat incremental.
The paper heavily relies on pre-existing systems, see [12] for the word embedding or [11] for the visual model.
The novelty seems to be in mapping a visual representation into a word embeddings by adding an intermediate layer.
However, this was proposed for instance in [15].
Of course, there are differences between papers 1048 and [15] as outlined lines 81-85:
* [15] reports results at a smaller scale.
* [15] uses a quadratic loss for the mapping while paper 1048 uses a rank loss.
* [15] does not use back-propagation to retrain the visual model.
But, again, these sound like incremental contributions.

- The design choices of the mapping layer seem ad-hoc and are poorly justified.
* It is stated that 'the embedding vectors learned by the language model are unit normed' (lines 159).
Is there any justification for such a nornalization?
* It is stated (lines 166-170) that 'training the model to minimize mean-squared difference ... produced poor results. We achieved much better results
with a combination of dotproduct similarity and hinge rank loss'.
Is there any justification for this besides 'it works better'?
* Similarly, several 'tricks' are proposed lines 179-182 and no justification is provided.
Why setting the margin to 0.1 for instance?

Here are additional comments/questions:

- There are many missing references on zero-shot recognition.
One of the most relevant ones is the following:
Palatucci, Pomerleau, Hinton, Mitchell, 'Zero-shot learning with semantic output codes', NIPS, 2009.
Especially, see section 4.1: words are embedded in a Euclidean space using a text corpus
and the a mapping is learned between inputs and word embeddings.
While the embedding is certainly much cruder than the one used in paper 1048, I believe this work is worth mentioning.

- Do you have any results in the case where there is no back-propagation into the core visual model?
Quantifying the impact of the back-propagation would be interesting.
Summary: While paper 1048 presents impressive results for large-scale zero-shot visual recognition,
its technical contribution is somewhat incremental as it looks like a combination of [11,12,15].

Submitted by Assigned_Reviewer_9

Summary of paper: This computer vision paper uses an unsupervised, neural net based semantic embedding of a Wikipaedia text corpus trained using skip-gram coding to enhance the performance of the Krizhevsky et al deep network [11] that won the 2012 ImageNet large scale visual recognition challenge, particularly for zero-shot learning problems (i.e. previously unseen classes with some similarity to previously seen ones). The two networks are trained separately, then the output layer of [11] is replaced with a linear mapping to the semantic text representation and re-trained on ImageNet 1k using a dot product loss reminiscent of a structured output SVM one. The text representation is not currently re-trained. The model is tested on ImageNet 1k and 21k. With the semantic embedding output it does not quite manage to reproduce the ImageNet 1k flat-class hit rates of the original softmax-output model, but it does better than the original on hierarchical-class hit rates and on previously unseen classes from ImageNet 21k. For unseen classes, the improvements are modest in absolute terms (albeit somewhat larger in relative ones).

Review: I think this is an interesting paper that can be accepted. The subject area (scaling to very large numbers of visual classes, combining modalities, deep learning) is clearly topical. This is not the first method to combine visual and textual information, but AFAIK it is the first to tackle this problem at such a large scale (all 21k ImageNet classes), especially using a fully unsupervised text model. The absolute improvements in recognition rates are fairly modest, but this is a challenging area where any improvement is welcome.

Further points

I have several questions for the authors:

- The skip-gram textual model is quite weak in the sense that no prior semantics is built in. It seems plausible that strengthening the semantic cues (e.g. with a WordNet distance based regularizer) would improve the embedding. Keying on Wikipaedia structures might also help, e.g. a "blue shark" might have its own page, with links back to the generic "shark" page.

- It would be useful to quantify the improvements available by refining the textual embedding during the joint training phase.

- How do adjectives / visual attributes come into this? Many classes have descriptive names, e.g. one might expect a "whitecap shark" to have some body part with a white cap. It is not clear to me that the current embedding model exploits such hints / such factorization in any usable way (e.g. as a "white" basis vector to be mixed in to the "shark" dot product).

- What are the flat precision scores for zero-shot DeVISE? - I ask because the hierarchical metric necessarily confuses the issue, especially at such low absolute precisions. E.g. if images of blue sharks are classified mainly as sharks, but not particularly as blue ones, the hierarchical score could still be quite high but one would hesitate to claim that a blue shark classifier has been learned. I suspect that this is happening here, at least to some extent.

Note added after rebuttal:

Excuse me for my garbled question about flat hit@k precision scores for zero shot DeVISE. I realise that you have already given these. What I meant to ask for was scores for a *hierarchical version* of the flat hit@k metric. Flat hit@k is the familiar "best of k" metric -- only the best of the k guesses counts towards the score. Your hierarchical precision@k is interesting but very different: it essentially measures how well the full set of k guesses reproduces the local categorical neighbourhood of the one true class. As such it is less robust than a "best of k" metric: even if a method invariably determines the true class with one of its k guesses, it will score badly on hierarchical precision@k if its other k-1 guesses are typically far from the true class. Given that you report the two metrics face to face and don't fully explain hierarchical precision@k in the paper, unwary readers are likely to think that hierarchical precision@k is a hierarchical "best of k" metric, and hence be mislead by the results. Also, to get a handle on the types of errors that are being made, it would be convenient to report at least some results in a metric that supports a "hops from ground truth" score decompositon. E.g. with the hierarchical best of k metric that counts how many hops from the true class the best [*] of the k guesses is, you can report accuracies as (say) 10% hits at zero hops (the flat hit@k score) plus 5% hits at one hop plus ... If desired this can be refined by breaking hop ties, e.g. counting children as closer than parents you might get "plus 3% hits to a child class plus 2% hits to the parent class plus...".

[*] Only one of the k guesses ever counts. Ties are broken arbitrarily.

Further points:

- Please use a uniform reporting convention for all performance score tables -- convert either Table 3 to percentages, or Table 2 to probabilities.


Summary: A decent paper on using text based semantic embedding to improve the performance of deep network classifiers for large numbers of visual classes, especially previously unseen ones. The actual experimental improvements are modest but the model is interesting and it is ambitious and topical to tackle the scaling to all 21k ImageNet classes.
Author Feedback

Author rebuttal: We’d like to thank all reviewers for their comments. Our responses to their concerns are below

- Several reviewers felt the paper is somewhat incremental, based on contributions [11], [12], and [15].
While it's true that our model is based on pre-existing building blocks, its integration is novel and achieves state-of-the-art performance on zero-shot prediction, while maintaining state-of-the-art performance on known classes, with semantically meaningful predictions in both cases. Furthermore, the proposed framework does not rely on any given ontology and would thus scale very well to much bigger corpora.

At a very high level, our work is algorithmically similar to [15]. The results however are very different. The approach in [15] works poorly on 8 classes (as evidenced by their use of a separate model to handle the known classes), and just satisfactorily on 2 novel classes. Our approach gives state-of-the-art performance on 1000 native classes with no side model, and generalizes well to 20,000 novel classes -- a scale of zero-shot learning two orders of magnitude larger than has even been attempted in the literature, and 1000x larger than [15]. Moreover, we believe that [15] optimized the wrong objective function (see below).

R7:

- Some design choices are poorly justified. In particular:

+ Why not mean squared error? We will make this point more clearly in the revision.
It’s not just that a ranking loss works better in practice, it’s that mean-squared difference does not capture the problem statement: The ultimate use of the system will be to emit an ordered list of predictions. Fundamentally this is a ranking problem and is best solved using a ranking loss. A mean-squared difference loss tries only to make the response close to the correct label, neglecting whether there are incorrect labels that are closer still. We tried mean-squared loss, and it halved the classification accuracy.
+ Why are the embedding vectors unit normed and why do we use a margin of 0.1?
Reference [12], which introduced the skip-gram model for text, used cosine distance between word vectors as their measure of semantic similarity (the norm of the vector being related to word frequency in the text corpus). Following their logic, we unit normed the vectors. The margin was chosen to be a fraction of the norm (1), and a wide range of values would likely work well.

- Missing references on zero-shot recognition, including Palatucci et al, NIPS 2009.
We will include this reference in the revision.

- Results with no back-propagation into the core visual model?
Back-propagation into the visual model provides a small improvement, typically in the range of 1-3% absolute over the model without back-propagation.

R8:

- The proposed model does not show any improvement over the baseline for the flat metric, only for the semantic metric.
Classification accuracy is one important measure in this domain, and as the reviewer points out, our model neither improves upon nor loses ground on this measure. But failing gracefully is a critical property for real world systems. And artificial visual system that misidentifies a telephoto lens as an eyepiece of some sort is strictly more intelligent that one that thinks it’s typewriter (real examples from the paper). The precision@k shows just this, and we believe is at least as relevant to real world uses cases as the flat classification accuracy measure.

- In zero-shot learning experiments, the proposed method is not compared with the state-of-the-art [A].
We should have cited [A] and compared our results to theirs, in the revision we will do both. In the short time allowed for rebuttal, we were not able to obtain and retrain on the exact train/test split used by [A], but we will do so in time for the revision. As a proxyl, we did a zero-shot experiment using the model trained on ImageNet 1K and tested using 250 classes randomly chosen from our 3-hop zero-shot data set, which maintains the same 80/20 split used by [A]. We ran two random selections of the 250 classes. For 250-way classification, our hit-at-5 accuracy is 31.5-35.7%, which matches the performance in [A]. For 1250-way classification, our hit-at-5 accuracy is 8.6-10.4%, which compares very favorably to the ~3% accuracy reported on the 1000-way classification graph in Fig 2 of [A]. Note that [A] requires a hand-engineered hierarchy to perform zero-shot, whereas our method needs only an unlabeled text corpus. Also note that our model performs better on the non-zero-shot flat metric (Table 1 in [A]).

- The implementation details are scarce and not sufficient to reproduce the results.
We thank the reviewers for pointing out a few details and hyperparameters that were missed in the paper. The final version will contain all the details needed to reproduce the results, with proper justification.

- In Supp. Material the validation subset of ImageNet 21K is used for zero-shot experiments. How exactly is it used?
The text in the Appendix was incorrect---there is no validation set from the ImageNet 21K used in zero-shot learning, the ImageNet 21K is only used for testing in that scenario. The mistake will be fixed in the final revision.

R9:

- The skip-gram textual model is quite weak.
We agree with the reviewer that model is fairly weak, but chose to use it because we were impressed with how much semantic information it gleaned in unsupervised training. Adding structure from WordNet might give gains in the ImageNet challenge, but is less scalable and maintainable than learning the semantics directly from text. Human curated knowledge representations are costly to scale, to maintain, and to keep current. For example, WordNet doesn’t contain the term “iPhone” whereas our model correctly learns that the most semantically related terms are “iPad”, “iOS” and “smartphone”.

- What are the flat precision scores for zero-shot DeVISE?
They are given in Table 2.